# Transcriptome Analysis Reveals Key Genes Involved in Trichome Formation in Pepper *(Capsicum annuum* L.)

**DOI:** 10.3390/plants13081090

**Published:** 2024-04-13

**Authors:** Yiyu Shen, Lianzhen Mao, Yao Zhou, Ying Sun, Junheng Lv, Minghua Deng, Zhoubin Liu, Bozhi Yang

**Affiliations:** 1Engineering Research Center of Education Ministry for Germplasm Innovation and Breeding New Varieties of Horticultural Crops, Key Laboratory of Vegetable Biology of Hunan Province, College of Horticulture, Hunan Agricultural University, Changsha 410128, China; bibisoli@163.com (Y.S.); maolz77@163.com (L.M.); y17_y11@163.com (Y.Z.); sy051844@163.com (Y.S.); 2College of Landscape and Horticulture, Yunnan Agricultural University, Kunming 650201, China; junhenglv@hnu.edu.cn (J.L.); dengminghua2013@sina.com (M.D.)

**Keywords:** pepper, trichome, transcriptome, gene

## Abstract

Trichomes are specialized organs located in the plant epidermis that play important defense roles against biotic and abiotic stresses. However, the mechanisms regulating the development of pepper epidermal trichomes and the related regulatory genes at the molecular level are not clear. Therefore, we performed transcriptome analyses of A114 (less trichome) and A115 (more trichome) to dig deeper into the genes involved in the regulatory mechanisms of epidermal trichome development in peppers. In this study, the epidermal trichome density of A115 was found to be higher by phenotypic observation and was highest in the leaves at the flowering stage. A total of 39,261 genes were quantified by RNA-Seq, including 11,939 genes not annotated in the previous genome analysis and 18,833 differentially expressed genes. Based on KEGG functional enrichment, it was found that DEGs were mainly concentrated in three pathways: plant–pathogen interaction, MAPK signaling pathway-plant, and plant hormone signal transduction. We further screened the DEGs associated with the development of epidermal trichomes in peppers, and the expression of the plant signaling genes *GID1B-like* (Capana03g003488) and *PR-6* (Capana09g001847), the transcription factors *MYB108* (Capana05g002225) and *ABR1-like* (Capana04g001261), and the plant resistance genes *PGIP-like* (Capana09g002077) and *At5g49770* (Capana08g001721) in the DEGs were higher at A115 compared to A114, and were highly expressed in leaves at the flowering stage. In addition, based on the WGCNA results and the establishment of co-expression networks showed that the above genes were highly positively correlated with each other. The transcriptomic data and analysis of this study provide a basis for the study of the regulatory mechanisms of pepper epidermal trichomes.

## 1. Introduction

Pepper (*Capsicum annuum* L.) as a crop of the genus *Capsicum* in the family Solanaceae, is an agricultural product with high economic value and great food value in China. Currently, crop yield problems caused by plant diseases and insect pests are one of the most important reasons plaguing the production of peppers [1]. Plant diseases and insect pests in crops are still controlled by spraying pesticides and other measures, which are extremely harmful to the environment and costly to produce, but it is of great economic value to cultivate highly resistant varieties and to improve the resistance of crop varieties to pests and diseases themselves [2,3]. The epidermal trichome structure of plants consists of specialized multicellular structures with the ability to synthesize and secrete secondary metabolites, which are often considered as the first line of defense to protect the plant from biotic and abiotic stresses such as resistance to insect pests, drought and water retention, salt tolerance, disease resistance, and defense against ultraviolet radiation and bright light damage [4,5,6,7]. The epidermal trichomes of peppers are glandular trichomes, and it has been shown that the distribution of pepper main stem and leaf tomentum is thought to be related to resistance silverleaf whitefly, blight, and mottle virus [8,9,10]. Trichome structure in plants is influenced and regulated by a variety of factors, including the physiological environment, phytohormones, regulatory genes, and miRNA [11].

The molecular mechanisms underlying the development of epidermal trichomes have been well characterized in single cells, especially in *Arabidopsis*, where it has been found that the development of trichomes is controlled by a number of different genes. Currently, the GL1-GL3/EGL3-TTG1 trichome developmental complex has been intensively studied, and a series of genes regulating the phenotypes of the number of epidermal trichome initiation, apical shape, organ distribution, and the number of branches have been identified or cloned, such as the genes required for the growth of epidermal trichomes, namely *TTG1*, *GL1*, *GL2*, *GL3*, *EGL3*, *GIS*, and *GIS2*, and genes that inhibit trichome growth, namely *TRY*, *CPC*, *CPC1 (ETC1)*, *ETC2*, *ETC3*, *TCL1*, *SPL*, and others [12,13,14,15,16]. The developmental mechanisms of epidermal trichomes in multicellular plants are still more limited compared to unicellular ones. Currently, studies on the epidermal trichomes of Solanaceae have focused on tomato, which has seven different morphologies (types I-VII) [17]. Tomato type I trichome formation is controlled by *Woolly (Wo)*, encoding the HD-ZIP IV transcription factor, and its interactors *SlCycB2* and *Hair (H)* [18]. Meanwhile, the study of Gao et al. [19] also demonstrated that the inhibition of *SlCycB2* expression suppresses the formation of type I trichomes. In addition, several other regulators of epidermal trichome regulation were identified, such as the *SlSRA1* tomato mutant showing trichome defects, and plants suppressed by *SlIAA15* and *SlARF3* showing a significant reduction in type I, V, and VI trichome formation [20,21,22]. Recent studies by Hua et al. [23] showed that overexpression of *SIJAZ4*, a negative regulator of JA signaling, significantly reduced the length of tomato trichomes, and Gong et al. [24] found that *SIMYB75* was able to participate in the formation of tomato trichomes and terpenoid synthesis through multiple regulatory pathways, which in turn affected the tolerance of plants to spider mites. These studies also provide a reference for the mechanism of epidermal trichome trait regulation in pepper. The genus Capsicum produces six major trichome types (types I-VI) [25], and Chunthawodtiporn et al. [26] identified two candidate genes controlling trichome formation on chromosome 10 based on physical location on the CM334 reference set of genomes through a population of high-generation autografts from a cross between the bell pepper type capsicums, ‘Maor’ and ‘CM334’, and were annotated to be TRICHOME BIREFRINGENCE-LIKE 5 (TBL5) and GLABROUS INFLORESCENCE STEMS (GIS), respectively. Liu et al. [27] identified a strong candidate gene, *CA10g21340*, associated with pepper epidermal trichome using two F_2_ populations, 18C2480 and 19Q6090, from the crosses of 18C2458 (Hairless) × 18C3375 (Hairy) and 19Q6092 (Hairy) × 19C6093 (Hairless), and found that the gene was highly homologous to *GIS* (AT3G58070) from *Arabidopsis* and to *H* (Solyc10g078970) from tomato.

Compared with other species, the epidermal trichomes of peppers have been less studied, the genes regulating trichome development have not yet been cloned, and the developmental regulatory mechanism of trichomes is still unclear, which limits the breeding process of peppers for resistance to plant diseases and insect pests. Therefore, the identification of key genes regulating the formation and development of epidermal trichomes in peppers and the analysis of their molecular regulatory mechanisms will provide a greater help to improve the resistance of pepper varieties to diseases and insect pests and reduce the use of chemicals in pepper cultivation. In this study, we selected stems and leaves of A114 (less trichome) and A115 (more trichome) plants at the seedling and flowering stages, conducted trait investigations and further transcriptome analyses, and screened and identified genes related to the development of epidermal trichomes of peppers using illumina sequencing and identified a series of related candidate genes. This study lays the foundation for further in-depth understanding of the regulatory mechanisms of pepper epidermal trichomes as well as directed breeding for improved disease resistance in pepper varieties.

## 2. Results

### 2.1. Phenotypic Observations on the Epidermal Trichomes of Peppers

In the present study, pepper variety A114 was almost completely free of trichome attachment on the epidermis of stems and leaves at the seedling and flowering stages. In contrast, in A115 the stems at the seedling stage were unevenly covered with trichomes, and trichome attachment on the leaves was sporadic from the third leaf of the plant (Figure 1a). At the flowering stage, the differences in epidermal trichome attachment between the two varieties were more pronounced, with a significantly higher density of trichomes on stems and leaves than at the seedling stage (Figure 1b). In addition, we selected the stems of A114 and A115 at the seedling stage for further observation using a microscope, and it could be found that the density of attached trichomes differed significantly between the two varieties of stems, and the trichomes on the stems of A115 were longer and denser compared with those of A114 (Figure 1c).

### 2.2. Illumina-Based Transcriptome Analysis and Identification of DEGs

A total of 175.93 Gb of Clean Data were obtained by Illumina sequencing, utilizing total RNA from eight groups and constructing cDNA libraries, with 6.27 Gb of Clean Data for each sample, and the percentage of Q30 bases was at 94.14% and above, which demonstrated the high throughput and high quality of the RNA-Seq data. After removing joints and low-quality sequences, a total of 589,624,675 clean reads were obtained, and these clean reads were compared with the *C. annuum L*_Zunla-1 database (https://www.ncbi.nlm.nih.gov/genome/10896, (accessed on 19 September 2021)), and the comparison rate was in the range of 85.16–94.93%. The eight groups were CL114 (leaves of A114 at the flowering stage), CS114 (stems of A114 at the flowering stage), ML114 (leaves of A114 at the seedling stage), MS114 (stems of A114 at the seedling stage), CL115 (leaves of A115 at the flowering stage), CS115 (stems of A115 at the flowering stage), ML115 (leaves of A115 at the seedling stage), and MS115 (stems of A115 at the seedling stage), and three replicates were performed for each sample. A total of 39,261 genes, including 11,939 genes not annotated in the previous genome analysis, were identified in 24 pepper samples (Appendix A). Further characterization revealed that 35,497 and 35,777 single genes were quantified in trichomeless pepper A114 and trichome pepper A115, respectively, and the total number of genes in both was 33,455. Among the four groups of trichome pepper A115, 403, 561, 639 and 777 genes were specifically quantified in CL115, CS115, ML115, and MS115, while 27,906 genes were co-quantified (Figure 2a,b).

We further identified 39,261 single genes, of which 18,833 were differentially expressed in the comparisons and contained 4902 new genes (Appendix A). Further analysis of the differential genes and up-regulated and down-regulated genes in the comparisons revealed that the number of differential genes in the leaves at the flowering stage was significantly higher in the trichomeless pepper A114 and the trichome pepper A115 than in the other comparisons, and that the same difference also existed in the comparisons for the leaves at the flowering stage and the leaves at the seedling stage across the comparisons of A115, suggesting that genes involved in the formation of pepper trichomes may be in the flowering stage leaves of A115 (Figure 2c–e). Meanwhile, screening for homologous genes related to epidermal trichome formation in DEGs revealed that the expression of *CaTBL5* (Capana10g002188) in A115 was significantly lower than that in A114 at all times. The differences in *CaGIS* (Capana10g002181, Capana10g002182) were more pronounced in leaves at the flowering stage and the expression was significantly lower in A115 leaves than in A114 leaves, while there were no significant differences in other periods. In addition, the expression of *CaCycB2* (Capana10g002051) was also significantly lower than that of A114 leaves in A115 leaves at flowering stage, but its expression was the opposite in other periods, especially in the stems at flowering stage where the expression of A115 was significantly higher than that of A114. The difference in the expression of *CaCycB3* (Capana06g000649) in the stems at flowering stage between the two materials was more significant, while there was no significant change in other periods.

### 2.3. GO Enrichment Analysis of DEGs

Gene ontology (GO) analysis was performed to explore the biological functions of DEGs in A114 and A115. In the comparison groups, DEGs were annotated into three major categories of GO terms, biological process (BP), cellular component (CC), and molecular function (MF). Using the comparator group CL114/CL115 as the filtering condition, the top 10 terms with higher enrichment content of DEGs in the three functional groups were selected and analyzed, and it was found that eight comparator groups had higher enrichment content of DEGs in the CC categories of membrane, membrane part, cell part, and cell, as well as in the MF categories of catalytic activity and binding, as well as higher enrichment of metabolic process, cellular process and single-organism process in BP. The above GO terms may play an important role in the growth and development of pepper trichomes. In addition, among the four groups compared in A114 and A115, DEGs enriched in the CL114/CL115 group were significantly higher than those in the other three groups (Figure 3a). To explore possible genes controlling trichome development in pepper, we screened three GO terms for three terms related to trichome development, namely trichome differentiation (GO: 0010026), trichome morphogenesis (GO: 0010090), and trichome branching (GO: 0010091); it was found that *ABIL3* (Capana03g001989) was up-regulated in both A114 and A115 comparator groups, and the difference was more pronounced in the CL114/CL115 group.

### 2.4. KEGG Pathway Analysis of DEGs

To further explore the functional network of biological interactions, KEGG pathway analysis was performed on DEGs. In this study, we used the comparison groups CL114/CL115 and CL115/CS115 as the screening conditions, and selected the KEGG pathways that were enriched with a high number of the top 20 DEGs for analysis, and found that the DEGs of the eight comparison groups were mainly enriched in plant–pathogen interaction, MAPK signaling pathway-plant, and plant hormone signal transduction pathways, which according to the results may be the three key KEGG pathways affecting the development of pepper trichomes. In the comparison of A114 and A115 at each site in each period, we analyzed and found that the number of DEGs enriched in each KEGG pathway in the CL114/CL115 group was significantly more than in the other comparative groups. Meanwhile, in the comparative group of A115, the DEGs enriched in the CL115/ML115 and CL115/CS115 groups were significantly more than in the other two groups (Figure 3b). In addition, our analysis revealed that photosynthesis, which was enriched in higher numbers of DEGs in each of the comparison groups of A114 and A115 and to a greater extent in the CL114/CL115 group, was not found in the first 20 pathways enriched in higher numbers in A115, suggesting that the trichomes of peppers may affect photosynthesis in the plant, and this was particularly obvious.

### 2.5. Analysis of DEGs Associated with Plant Hormone Signal Transduction Pathways

In the above KEGG pathway analysis, it was found that DEGs in each comparison group were mainly enriched in plant hormone signal transduction pathway, and it was hypothesized that this pathway plays a key role in trichome development. Eight plant signaling pathways were detected in this study, namely IAA, ETH, CTK, GA, SA, JA, ABA, and BR. Comparative analyses of A114 and A115 revealed that a part of genes differed significantly in each comparison group of A114 and A115, and their expression was at a higher level in the leaves at the flowering stage of A115, such as the expression of *GID1B* (Capana03g003488) in gibberellin (GA) signaling pathway, *ERF01B-like* (Capana05g001701) in ETH, *PR-6* (Capana09g001847) and *PR1-like* (Capana08g002192) in SA, *SERK2-like* (Capana01g001931, Capsicum_ new14288) and *BRI1* (Capana06g000920) in BR, and *PG-like* (Capana09g000144) in JA. In addition, we found that some components showed the same expression pattern in the hormone signal transduction pathway. The gene expression of GID1, PR-1, and ABF were all significantly up-regulated in A115 material compared to A114. In ETH, the expression of SIMKK was significantly higher in stems than in leaves, and the expression of most DEGs was higher in A114 than in the same part of A115 at the same time. Based on the expression analysis, the above phytohormone-regulated genes may be involved in the regulatory mechanism of pepper epidermal trichomes (Figure 4).

### 2.6. Analysis of Transcription Factors Associated with Trichome Development

Many transcription factors are involved in plant hormone regulation as well as in the formation and development of plant trichomes. To further elucidate the potential roles of transcription factors in regulating trichome development in pepper, a total of 17 transcription factor families were detected in the pepper samples, in descending order according to the number of genes they contained: MYB, AP2/ERF, bHLH, WRKY, NAC, HD-ZIP, DOF, B3, TCP, GATA, HSF, CCCH, NF-Y, bZIP, Trihelix, WD40, and C2H2. We analyzed the MYB and AP2/ERF transcription factor families with a high number of genes using the identification of FPKM > 10 in at least one sample as a screening condition (Appendix A, Figure 5a).

In trichome material A115, the MYB transcription factor genes *MYB48* (Capana11g000757), *MYB48-like* (Capana06g002789), and *MYB330-like* (Capana00g004862) were more highly expressed than A114 at all times, and *DIV-like* (Capana09g001568) showed the opposite pattern. In addition, the gene expression of *MYB4* (Capana03g003830) and *MYB340-like* (Capana03g000696) was significantly lower at A115 than at A114 at the seedling stage, and vice versa at the flowering stage. Most of the genes in the AP2/ERF family of transcription factors showed a lower gene expression at A115 than at A114 in comparison with the seedling stage of the plants, and vice versa at the flowering stage, including *RAV1* (Capana11g002234), *RAP2-13* (Capana04g001107), and *ERF05* (Capana05g001951, Capana05g001949, Capana12g002081). There were also genes with higher gene expression at both seedling and flowering stages in A115 than in A114, including *ERF010-like* (Capana04g000577), *ERF01B-like* (Capana05g001701), and *ERF01B-like* (Capana02g000496). It was also found that *ERF02-like* (Capana01g000677), *ERF054* (Capana00g004561), and *ABR1-like* (Capana04g001261) were expressed significantly higher than the other groups in A115 leaves at the flowering stage (Figure 5d). Based on these results, it is suggested that the above transcription factors may be involved in the development of pepper trichomes, especially expressed in pepper flowering stage leaves.

### 2.7. Analysis of DEGs Associated with Plant Resistance

The trichome structure of plants plays a crucial role in protecting plants from biotic and abiotic stresses. In this study, we identified 907 plant resistance genes (PRGs), including 13 categories, which were, in descending order according to the number of genes they contained, RLK, RLP, L, NL, N, CNL, CN, RLK-GNK2, CN-R1, T, TNL, TN, and CNL-PRW8 (Appendix A, Figure 5b). Among them, in the comparison groups, there were a total of 522 PRGs differentially expressed. In the comparison groups of A114 and A115, there were 259, 142, 51, and 82 DEGs in CL114/CL115, CS114/CS115, ML114/ML115, and MS114/MS115, respectively, and the number of up-regulated genes was greater than the number of down-regulated genes in both stem leaves at the flowering stage as well as in leaves at the seedling stage (Figure 5c). We analyzed RLK and RLP, which had a higher number of genes, and found that for *At5g48380* (Capana06g000234), *At1g06840* (Capana09g000823), *At2g40270* (Capana06g002255), and *RLK5* (Capana09g000232) in RLK, as well as *SERK2-like* (Capana01g001931), *PGIP-like* (Capana09g002077), and *BRI1* (Capana06g000921) in RLP, the expression of the genes was higher in A115 than in A114 in the same part of the same period of time at both flowering stages A114 and A115, and the expression in the A115 expression was significantly higher in leaves at the flowering stage than in other periods (Figure 6c). According to these results, it was inferred that the formation and development of epidermal trichomes in peppers enhanced the resistance of the plant itself to a certain extent.

### 2.8. Analysis of Co-Expression Associated with Trichome Development

To further explore the regulatory mechanisms of trichome formation and development in pepper as well as the transcription factors and other related genes that may be involved, WGCNA was utilized to screen for important modules as well as to construct gene co-expression networks based on 18,833 differentially expressed genes and pepper-related phenotypes. In this study, a total of 17 different co-expression modules were generated and labeled with different colors, and we found that the top three modules with more enriched genes were turquoise, blue and brown, containing 4601, 2732, and 2312 genes, respectively (Figure 6a). Each module contained positively and negatively correlated genes, and 5 of the 17 co-expression modules were significantly correlated with traits, with the blue module showing a very high positive correlation with the CL115 group (r = 0.95, *p*-value = 3 × 10^−4^) (Figure 6b). Combined with the above analysis of GO, KEGG, and other results, it was hypothesized that the genes in the blue module might be involved in the formation and development of pepper trichomes, and 20 hub genes in the blue module were selected to construct a gene co-expression network map with the involved related transcription factors, phytohormone-related genes, and plant resistance genes (Figure 6c). The analysis revealed high positive correlations between most transcription factors, plant signaling genes, and plant resistance genes with hub genes. In addition, we found that the plant signaling gene *GID1B-like* (Capana03g003488) was positively correlated with the transcription factors *MYB108* (Capana05g002225) and *ABR-1* (Capana04g001261), the plant signaling genes *PR-6* (Capana09g001847) and *BRI1* (Capana06g000920), and the plant resistance genes *PGIP-like* (Capana09g002077) and *At5g49770* (Capana08g001721), which all showed high positive correlation. Taken together, these results suggest that the molecular regulatory mechanism for constructing the trichome phenotype of pepper is very complex, including different transcription factor families and phytohormone-related functional genes, etc., and that the above genes may have a critical role in trichome formation.

### 2.9. Analysis of Co-Expression Associated with Trichome Development

We randomly selected important genes related to trichome formation in peppers for RT-qPCR validation. The results showed (Figure 7) that the relative expression in leaves and stems of trichomeless pepper A114 and trichome pepper A115 at the flowering and seedling stages showed similar expression trends with the transcriptome sequencing results, indicating the reliability of the transcriptome sequencing data in this study.

## 3. Discussion

Plant epidermal trichomes improve plant resistance, and breeding pepper varieties with high resistance can improve pepper yield to some extent. However, the molecular regulatory mechanisms of epidermal trichome formation and development in pepper are still unclear. In this study, the material we used, A114, had sporadic short trichomes on the stems and leaves and exhibited trichome-free morphological characteristics. The other pepper variety, A115, had a high density of long trichomes as well as some short trichomes on the stem and leaves of the plant, exhibiting trichome characteristics. The characterized morphology of these two materials is an ideal candidate for studying the developmental regulatory mechanism of epidermal trichomes in peppers, making the subsequent analysis data and results more informative.

Previous studies have shown that trichome formation and development are influenced by hormone signaling and play a crucial role in plant defense [11]. In a previous study on the epidermal trichome of peppers, Gao et al. [28] conducted a comparative transcriptome analysis of the peppers GZZY-23 (hairy) and PI246331 (hairless), and compared with the data analysis of the present study, we found the existence of similar expression, such as KEGG enrichment analysis as well as phytohormone signaling analysis. On the basis of this experiment, we added the comparison of pepper varieties at the six-leaf-one-center period and the flowering period, and the phenotype found that the difference in the density of epidermal trichome was more obvious at the flowering period, so that we could further target the potential genes that might affect the development of epidermal trichome in peppers based on the phenotypic results. In addition, we combined with the co-expression analysis to target the modules that were highly correlated with the phenotypic results, to improve the reliability of the data, and to speculate on the mechanism of the development of epidermal trichome formation in the pepper varieties. We reviewed Liu et al. [27] and screened the strong candidate gene *CA10g21340* related to velvet hairs and found that the expression of this gene was low in this study, and hypothesized that there might be differences in the regulatory genes for trichome morphology in different peppers, and this study provides other possible candidate genes for epidermal trichome formation. In the KEGG enrichment analysis of this study, we found that DEGs were mainly enriched in the plant hormone signal transduction pathway. Related studies have found that different plant hormone stimuli produce different types of trichome structures [29,30]. The gene regulatory mechanisms of epidermal trichomes in *Arabidopsis* have been intensively investigated in mechanistic models showing that gibberellins (GAs) and cytokinins (CTKs) affect the formation of epidermal trichomes in plants [31]. GAs regulates the development of epidermal trichomes in plants mainly through interactions with transcription factors [30]. In *Arabidopsis*, GA relies on the transcription factors GL1 and TTG to promote the development of epidermal trichomes [32]. The C2H2 zinc-finger transcription factor GIS controls the development of tobacco epidermal trichomes by regulating GA signaling [33,34]. Chen et al. [35] found that the content of GA was significantly reduced in tomato lines overexpressing *SlbHLH95*, a negative regulator of trichomes. The above studies suggest that GA is a major phytohormone in the development of plant epidermal trichomes. In this study, we found that most of the genes related to the GA signaling pathway were up-regulated in the comparison of trichomeless-pepper A114 and trichome-pepper A115, and *GID1B-like* (Capana03g003488) was abundantly expressed in the leaves of A115 at the flowering stage, and it is speculated that the GA signaling pathway may be involved and play an important role in the developmental mechanism of the epidermal trichomes of peppers. It is hypothesized that the GA signaling pathway may be involved in the developmental mechanism of pepper epidermal trichomes and play an important role in regulating the development of epidermal trichomes by interacting with transcription factors.

The formation and development of plant epidermal trichomes are influenced by regulatory genes. Several studies have shown that transcription factors are involved in the regulation of plant trichome development, including MYB, bHLH, HD-Zip, and AP2/ERF [11,36,37]. In the present study, we analyzed the MYB and AP2/ERF transcription factor families in further depth based on the results of the number of genes included in the transcription factor families. In *Arabidopsis*, mutations in the R2R3 MYB transcription factor gene *GL1* resulted in the formation of a glabrous phenotype in plants [38]. The functional homologue of *GL1*, *GhMYB2*, promotes trichome development in cotton [39]. Matías-Hernández et al. [40] found that both *AaMYB1* and its homologue, *AtMYB61*, affect terpene metabolism and the development of trichome development in *Artemisia annua* and *Arabidopsis*. In contrast, ectopic expression of the R3-MYB transcription factor gene *OsTCL1* in *Arabidopsis* affected epidermal trichome and root hair formation [41]. Gong et al. [24] found that *SIMYB75* is able to participate in tomato trichome formation and terpenoid synthesis through multiple regulatory pathways, which in turn affects plant tolerance to spider mites. The above studies indicate that the MYB family of transcription factors is widely involved in the formation and development of plant epidermal trichomes. In this study, the MYB transcription factor family had the highest gene content, and the expression of genes *MYB48* (Capana11g000757), *MYB48-like* (Capana06g002789), and *MYB4* (Capana03g003830) was found to be higher in trichome peppers A115 than A114 at the same time period, which, combined with the phenotypic observation speculated that the above MYB transcription factor family genes might be involved in the formation and development of epidermal trichomes in pepper with different regulatory mechanisms. In addition, a family of transcription factors with high gene content is AP2/ERF, which regulates plant development and is also involved in many phytohormone signaling pathways [42]. In rice, it was found that the AP2/ERF transcription factor, HL6, interacts with *OsWOX3B* to regulate the expression of the growth hormone-related gene, *OsYUCCA5*, which affects plant development of epidermal trichomes [43]. Wang et al. [44] found that overexpression of another gene of this family, *AaWIN1*, in *A. annua* significantly increased the density of epidermal trichomes, and ectopic expression in *Arabidopsis thaliana* affected the development of leaves and trichomes in the plant. *ERF01B-like* (Capana05g001701), *ERF01B-like* (Capana02g000496), *ERF02-like* (Capana01g000677), *ERF054* (Capana00g004561), and *ABR1-like* (Capana04g001261) were all higher in A115 than in A114 at the same time period. It was hypothesized that the above AP2/ERF transcription factor genes might be involved in the regulatory mechanism of the development of epidermal trichomes in peppers, or influence the formation of epidermal trichomes by affecting the phytohormone signaling pathway. In addition, related studies showed that the overexpression of *MYB4*, *ZmMYB48*, *PhERF2*, and *ABR1* enhanced plant resistance [45,46,47,48].

The epidermal trichomes of plants can form spatial barriers to defend against external biotic and abiotic stresses. Numerous studies have shown that plant resistance genes (PRGs) can enhance plant resistance to bacteria, fungi, and ototoxic memory viruses [49,50,51,52]. The largest class of PRGs encodes structural domains that contain leucine-rich repeats (LRR), which can be found in membrane-carrying proteins during protein–protein interactions, and are known as leucine-rich repeats (LRR-RLP) and LRR receptor-like kinases (LRR-RLK) [53,54,55]. LRR-RLP and LRR-RLK are key receptors on the cell surface that sense pathogen invasion to establish signaling circuits that regulate growth, development, and immune function in plants, among others [56,57,58,59]. The promoter of LRR-RLK promoters have cis-elements for phytohormone and stress response, which are involved in the brassinosteroid (BR) signaling pathway, and studies have shown that BR has great potential for crop trait improvement. BRI1 is a key LRR-RLK in the BR signaling pathway, and synergistically with BAK1, it forms the BRI1/BAK1 complex that affects BR signaling [60,61,62,63,64]. In addition, SERK2, another component of BR, also plays a role in resistance. In rice, overexpression of *SERK2* significantly enhanced the plant’s resistance to salt stress and to *Xanthomonas oryzae pv. oryzae (Xoo)*, and the highest expression of *SERK2* was detected in leaves based on expression [65,66]. At the same time, it was found that BAK1 plays an important role in the abscisic acid (ABA) signaling pathway. In *Arabidopsis*, this signaling pathway can be used to regulate plant stem elongation and fertility [67]. In this experiment, both plant resistance genes *At5g49770* (Capana08g001721) and *At1g06840* (Capana09g000823) were detected to belong to the LRR-RLK family in *Arabidopsis*.

## 4. Materials and Methods

### 4.1. Plant Material and Phenotypic Observations

In this study, A114 (less trichome) and A115 (more trichome) peppers were used, and both materials were provided by the Pepper Team of the College of Horticulture, Hunan Agricultural University. After germination, both materials were cultured in a uniform growth environment with a temperature range of 28/20 °C, a photon flux density of 200 ± 10 μmol·m^−2^·s^−1^, a light–dark cycle of 16/8 h, and a relative humidity of 65% ± 5%. Six-leaf one-heart stage plants (M) and flowering stage (C) plants with uniform growth were selected from both plant varieties, and their leaves (L) and stems (S) were collected for the study. A114 and A115 were divided into eight groups numbered as ML114, MS114, CL114, CS114, ML115, MS115, CL115, and CS115. Meanwhile, the selected A114 and A115 were observed under the electron microscope at the six-leaf one-heart period of peppers. Each subgroup was mixed and divided into three copies for RNA-Seq assay, and the rest were stored at −80 °C for subsequent experiments.

### 4.2. RNA Extraction, Library Construction, and Sequencing

Total RNA was extracted from leaves and stems of two materials, A114 and A115, according to the manufacturer’s instructions of TRIzol (Invitrogen, Carlsbad, CA, USA) and checked for completeness of RNA samples using 1% agarose gel electrophoresis and GelView 1500Plus (BLT Photon Technology, Guangzhou, China) to check the integrity of RNA samples to avoid contamination and degradation of extracted RNA for the construction of RNA sequencing libraries. For the obtained RNA samples, sequencing libraries were generated using the NEBNext UltraTM RNA Library Prep Kit (NEB, Ipswich, MA, USA) for Illumina. mRNA was enriched using magnetic beads and randomly fragmented into short fragments, which were used as templates to synthesize cDNA, purified, end-repaired, A-tailed, and ligated with sequencing junctions, and enriched by PCR to obtain cDNA libraries. The cDNA sequencing libraries were tested for purity and quality using an Agilent 2100 Bioanalyzer (Agilent Technologies, Inc., Santa Clara, CA, USA). According to the vendor instructions of Biomarker Technologies (Wuhan, China), different libraries were pooled according to the target downstream data volume, and 24 cDNA libraries were sequenced using the Illumina HiSeq 2500 platform.

### 4.3. RNA-Seq and Differential Expression Analysis

To ensure the accuracy of the subsequent analyses, the raw data after library sequencing were checked by quality testing to remove reads containing junctions and those of low quality (proportion of N > 10%, and reads with quality values Q ≤ 10 bases that accounted for more than 50% of the whole Read), and were calibrated for Q20 and Q30 base percent occupancy. The screened clean reads were compared with the reference genome *Capsicum annuum* L_Zunla-1 (https://www.ncbi.nlm.nih.gov/genome/10896, (accessed on 19 September 2021)) of pepper using TopHat 2.0 software. The contrasted reads were assembled and evaluated for expression using String Tie 1.3.3 software, and compared with the original genome annotation information to uncover unannotated transcribed regions and find new transcripts as well as new genes. Differential expression analysis between sample groups was calculated using DESeq2 R (v.1.30.1) software package, and fold change ≥ 2 and false discovery rate (FDR) < 0.01 were used as the screening criteria to identify differentially expressed genes between sample groups for further analysis.

### 4.4. GO and KEGG Pathway Analysis

To further characterize the gene functions of DEGs, the differentially expressed genes were functionally annotated in a database. The DEGs of the samples were classified and enriched using the GO Function database (http://www.geneontology.org/, (accessed on 20 September 2021)), and the number of genes associated with each GO Term was calculated; to analyze whether the differentially expressed genes differed significantly on a given pathway, an enrichment analysis was performed based on the KEGG pathway database (https://www.genome.jp/kegg/, (accessed on 20 September 2021)). FDR ≤ 0.05 was used as the significant enrichment threshold for correction.

### 4.5. Weighted Gene Co-Expression Network Analysis (WGCNA)

To understand the gene association patterns among different samples, co-expression networks were constructed by analyzing differentially expressed genes in A114 and A115 pepper at different periods using the WGCNA R package [68]. Co-expression gene modules were constructed using selected DEGs, and different genes were categorized into different co-expression modules using the dynamic tree-cutting method, with the different gene modules indicated by different colors, and based on the phenotypes of the gene modules with the samples and the endogeneity of the gene modules, key genes were identified for subsequent analyses. Modules with the phenotypes of the samples and the endogenous nature of the gene modules to identify the key genes for subsequent analysis. According to the connectivity of the genes in the modules, 20 genes with higher connectivity were screened as hub genes, and the regulatory relationships between the identified hub genes and other trait-related genes in the modules were visualized using Cytoscape 3.7 software to construct the correlation network diagram.

### 4.6. Quantitative RT-PCR Analysis

The relative expression levels of the screened DEGs were verified by RT-qPCR. Total RNA was extracted with SteadyPure Universal RNA Extraction Kit (Accurate Biotechnology (Hunan) Co., Ltd., Changsha, China), and cDNA was synthesized as template using cDNA Synthesis Kit (Vazyme, Jiangsu, China). Twelve related genes were selected in DEGs, and gene-specific primers for qPCR were designed according to the sequences selected in RNA-seq (Appendix A). The relative gene expression was normalized using the $2^{-\Delta \Delta C_{T}}$ method.

## 5. Conclusions

In this study, we performed a transcriptome analysis of two pepper varieties with and without trichomes to dig deeper into candidate genes that might be involved in the developmental regulatory mechanisms of pepper epidermal trichomes. We screened transcription factor families, genes involved in phytohormone regulation, and plant resistance genes, which may be regulatory DEGs involved in the developmental mechanism of pepper epidermal trichomes. Among them, we found that *GID1B-like* (Capana03g003488), which regulates plant GA signaling genes, was more highly expressed in pepper varieties with trichomes than in varieties without trichomes and was highly expressed in leaves at the flowering stage, which is more consistent with the results of trichome density observed phenotypically. And this gene has a high positive correlation with transcription factors *MYB108* (Capana05g002225) and *ABR-1* (Capana04g001261), as well as plant signaling genes *PR-6* (Capana09g001847) and *BRI1* (Capana06g000920). Therefore, we hypothesized that GID1B is involved in the developmental mechanism of pepper epidermal trichomes and may be regulated by MYB and AP2/ERF transcription factors or modulate transcription factors, which in turn affect the formation of pepper epidermal trichomes. In addition, we found that most of the screened transcription factors and phytohormone-regulated genes were associated with plant resistance, and that the formation of pepper epidermal trichomes contributes to the increase in the expression of plant resistance genes, which may be able to enhance the plant’s ability to withstand biotic and abiotic stresses. This study contributes to an in-depth understanding of the developmental mechanism of epidermal trichomes in peppers and provides valuable candidate genes for molecular breeding of pepper varieties for disease resistance.

## Figures and Tables

**Figure 1 plants-13-01090-f001:**
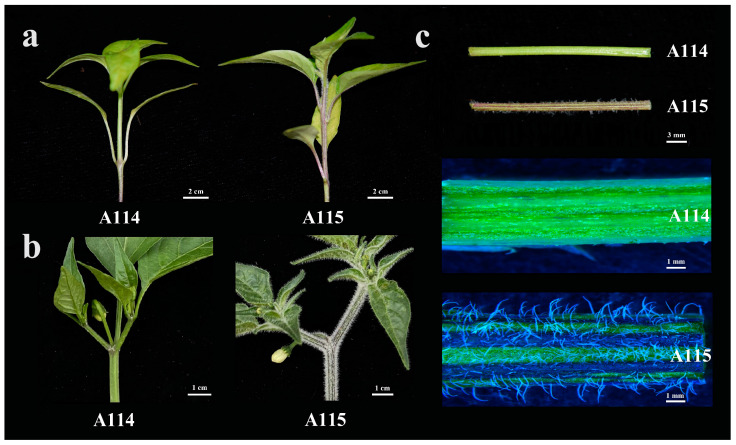
Phenotypes of pepper varieties A114 and A115. (**a**) A114 and A115 at the seedling stage. (**b**) A114 and A115 at the flowering stage. (**c**) Microscope scanning photographs of the stems of A114 and A115 at the seedling stage.

**Figure 2 plants-13-01090-f002:**
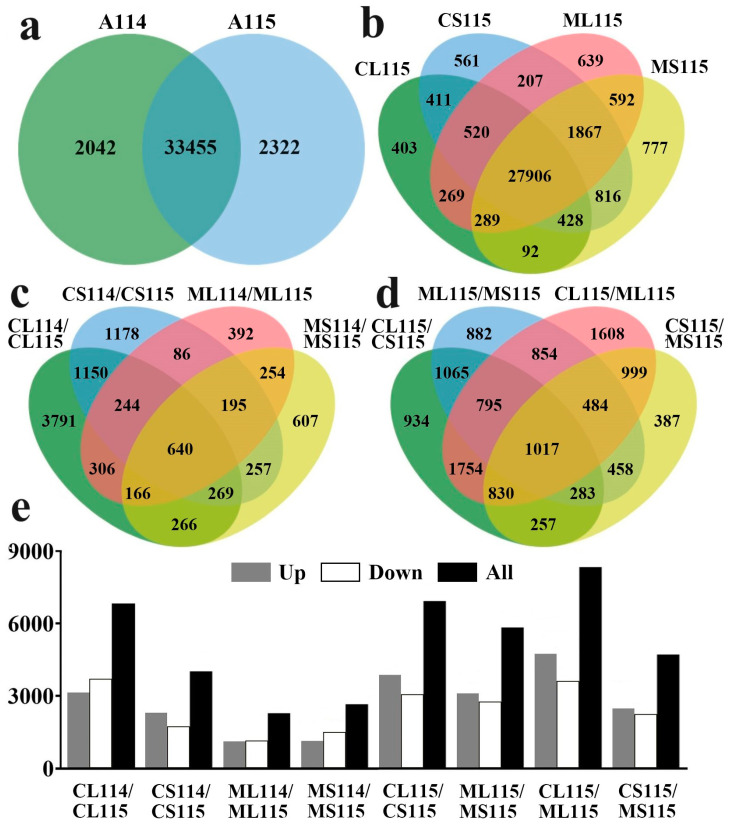
Statistical analysis of genes identified by transcriptome analysis. (**a**) Venn diagram of co-expressed and uniquely expressed genes in A114 and A115. (**b**) Venn diagrams of co-expressed and uniquely expressed genes at different times and organs in A115. CL, leaves at the flowering stage; CS, stems at the flowering stage; ML, leaves at the seedling stage; MS, stems at the seedling stage. (**c**,**d**) Venn diagram of co-expressed and uniquely expressed DEGs in the different comparison groups. (**e**) The number of DEGs as well as up-regulated and down-regulated genes between different comparison groups.

**Figure 3 plants-13-01090-f003:**
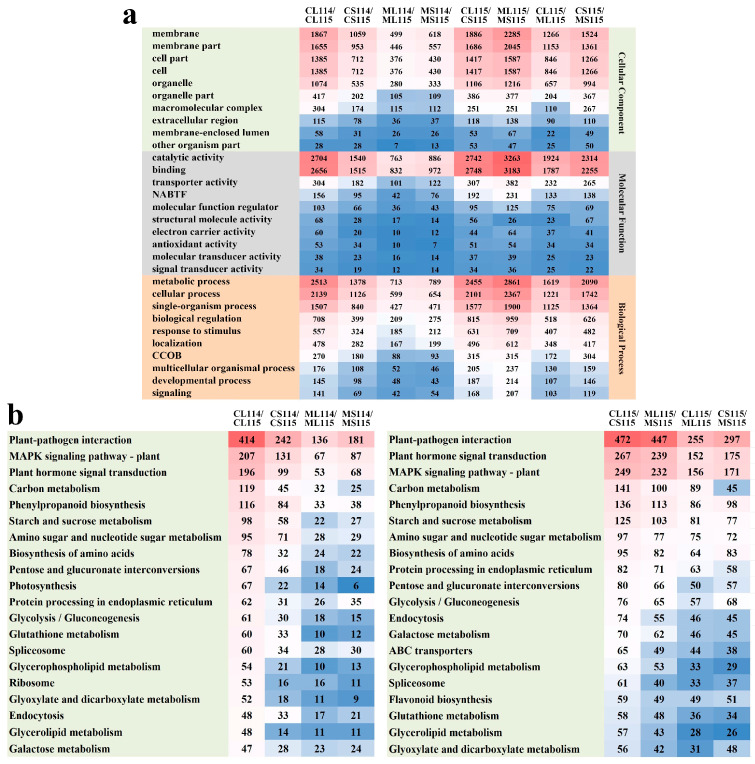
Enrichment analysis of DEGs in different comparison groups. CL, leaves at the flowering stage; CS, stems at the flowering stage; ML, leaves at the seedling stage; MS, stems at the seedling stage. (**a**) GO classification annotation map of DEGs. The top 10 GO terms with the highest enrichment content of DEGs in the three functional groups. NABTF, nucleic acid binding transcription factor activity; CCOB, cellular component organization or biogenesis. (**b**) KEGG pathway annotation map of DEGs. The top 20 KEGG pathways with the highest enrichment content of DEGs.

**Figure 4 plants-13-01090-f004:**
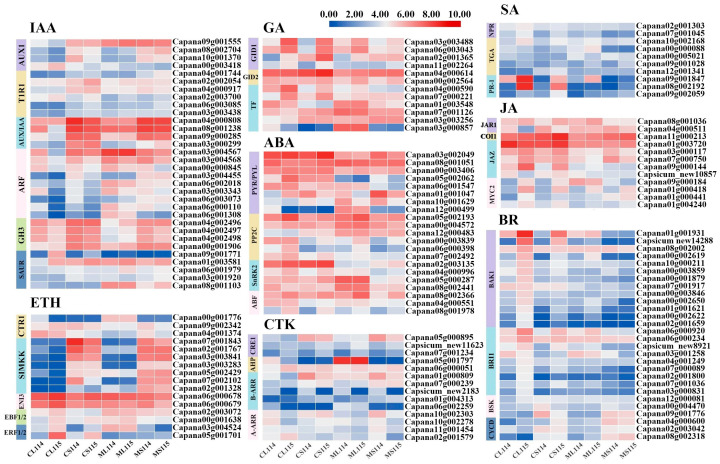
Heat map (log FPKM) of the expression of DEGs associated with plant hormone signal transduction pathways in A114 and A115. CL, leaves at the flowering stage; CS, stems at the flowering stage; ML, leaves at the seedling stage; MS, stems at the seedling stage.

**Figure 5 plants-13-01090-f005:**
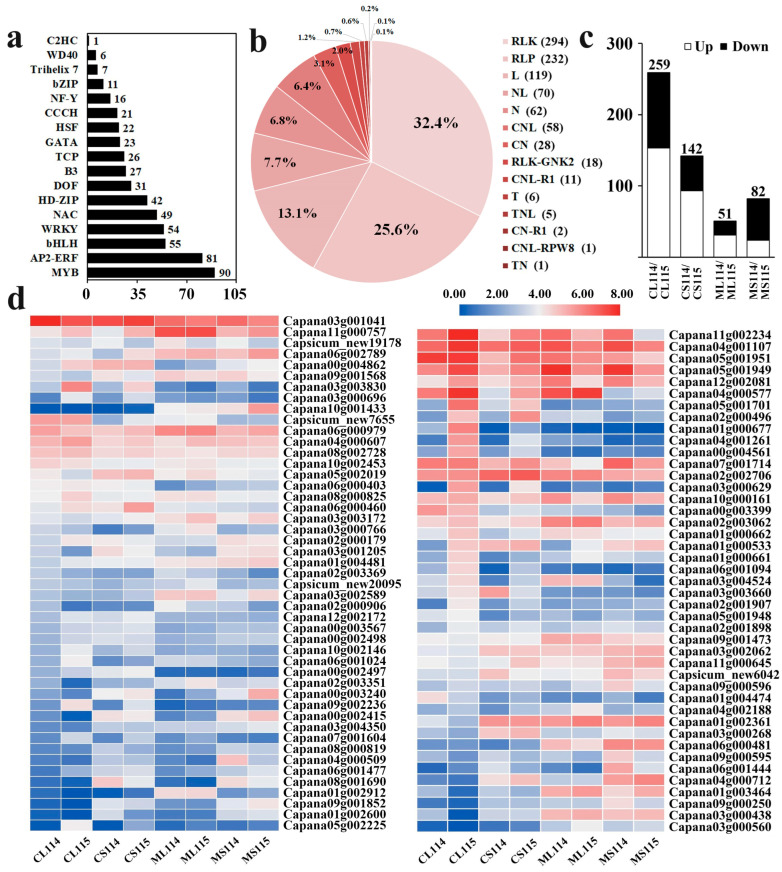
Analysis of plant resistance genes and transcription factors. CL, leaves at the flowering stage; CS, stems at the flowering stage; ML, leaves at the seedling stage; MS, stems at the seedling stage. (**a**) Number of associated transcription factor families. (**b**) Classes of plant resistance genes. (**c**) Number of up-regulated and down-regulated genes in each comparison group. (**d**) Heat map (log FPKM) of MYB and AP2/ERF transcription factor families gene expression.

**Figure 6 plants-13-01090-f006:**
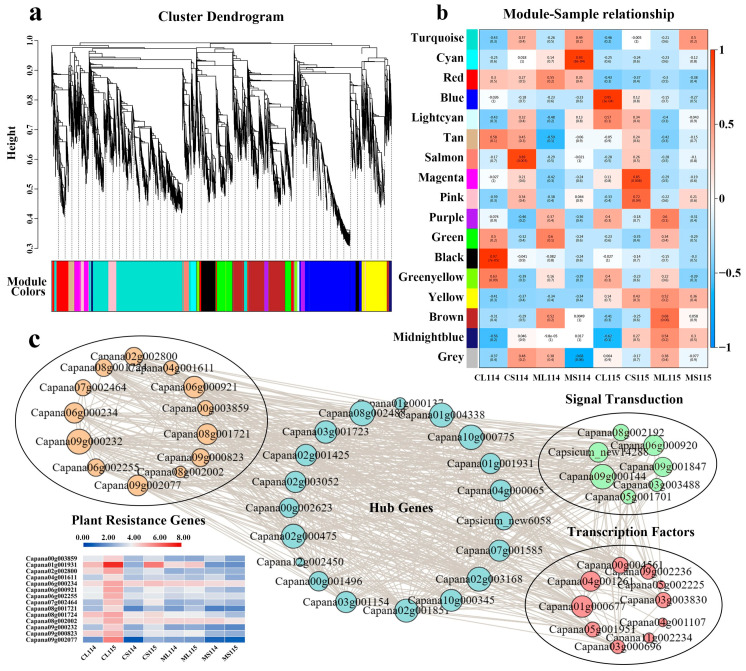
Co-expression network analysis of trichome development in peppers. CL, leaves at the flowering stage; CS, stems at the flowering stage; ML, leaves at the seedling stage; MS, stems at the seedling stage. (**a**) Module hierarchical clustering tree based on WGCNA analysis. (**b**) Relationships between modules and key traits in samples. (**c**) Co-expression network diagram of candidate genes in the blue module with genes related to trichome development. Solid line indicates a positive correlation between genes.

**Figure 7 plants-13-01090-f007:**
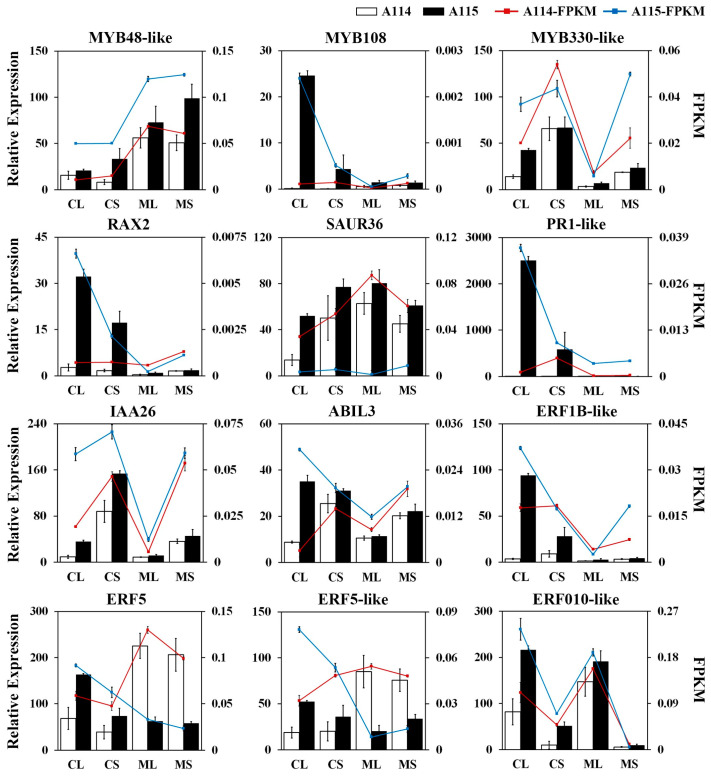
Validation and expression analysis of selected genes associated with trichome development using RT-qPCR. CL, leaves at the flowering stage; CS, stems at the flowering stage; ML, leaves at the seedling stage; MS, stems at the seedling stage. *MYB48-like*, Capana06g002789; *MYB108*, Capana05g002225; *MYB-330like*, Capana00g004862; *RAX2*, Capana09g002236; *PR1-like*, Capana08g002192; *SAUR36*, Capana01g003581; *IAA26*, Capana03g000244; *ABIL3*, Capana03g001989; *ERF1B-like*, Capana05g001701; *ERF5*, Capana05g001949; *ERF5-like*, Capana12g002081; *ERF010-like*, Capana04g000577.

## Data Availability

The data presented in this study are available on request from the corresponding author. RNA-Seq data generated in this study is available from the SRA-Archive (http://www.ncbi.nlm.nih.gov/sra, (accessed on 20 September 2021)) with accession number PRJNA1056125.

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
