# Peer review of "Transcriptome Analysis Reveals Key Genes Involved in Trichome Formation in Pepper (Capsicum annuum L.)"

_plants, 2024, doi:10.3390/plants13081090_

Round 1

Reviewer 1 Report

Comments and Suggestions for Authors

[General comments]

1. I believe that there needs to be proper citation and discussion based on several previous studies on trichome formation in Capsicum annuum. In particular, Gao et al. (2021) Sci Rep (https://www.nature.com/articles/s41598-021-89619-0) and Lie et al. (2021) Frontiers in Plant Sci (https://www.frontiersin.org/journals/plant-science/articles/10.3389/fpls.2021.784755/full) should be cited, compare their results with the results you have shown in this manuscript, and state in the discussion part what parts of your results are novel. This will dramatically increase the value of this manuscript.

2. Are there any physiological differences between A114 and A115? For example, differences in growth rate, disease resistance, etc. Are there any of these physiological differences that can be explained by this transcriptome analysis?

[Specific comments]

3. Figure 1: Please indicate the scale bars.

4. Figure 2-4, Figure 5 left panel: Please enlarge the figure a little larger so that the text can be read; it should be about the size of the text on the right panel of Figure 5d. 

5. L 226: hormone > plant hormone

6. L 441: Please show the name of the cDNA library preparation kit.

Author Response

Dear Editors and Reviewers:

Thank you for your letter and for the reviewers’ comments concerning our manuscript entitled “Transcriptome analysis reveals key genes involved in trichome formation in pepper (Capsicum annuum L.)”. Those comments are all valuable. We have studied the comments carefully and have made corrections which we hope meet with approval. Revised portions are marked in red on the manuscript.

Reviewer 2 Report

Comments and Suggestions for Authors

The authors investigated the regulatory mechanisms of epidermal trichome development in pepper using transcriptome analysis. They compared the trichome density and gene expression profiles of A114 (less trichome) and A115 (more trichome) pepper plants. The study identified differentially expressed genes (DEGs) associated with trichome development and highlighted pathways such as plant-pathogen interaction, MAPK signaling, and plant hormone signaling. Co-expression analysis further supported the regulatory roles of these genes in trichome development. The study advances our understanding of the molecular mechanisms underlying epidermal trichome development in pepper. In conclusion, the paper would be sufficient to merit publication in Plants, though a revision is recommended which needs to include the following points.

(1) Figure 1

Panel B should also be marked with A114 and A115.

(2) line 115: Please show the statistics of transcriptome data.

(3) line 117: Please explain what the sample name represents. At the same time, briefly describe the experimental design of the RNA-seq analysis. It is mentioned in the M & M section, but it is easier for the reader to understand if I explain it here.

(4) line 124: The author says "new genes," but I think that is not accurate. These genes were only unannotated in the previous genome analysis and should be described as such. 

(5) Fig. 4 and 5 Heatmap: Please explain on which values the heat map was drawn. logFPKM? 

(6) line 292, Fig6: Hug genes -> Hub genes? 

Author Response

(The authors gave the same response as above.)
